# Palliative and End-of-Life Care for People Living with Motor Neurone Disease: Ongoing Challenges and Necessity for Shifting Directions

**DOI:** 10.3390/brainsci13060920

**Published:** 2023-06-07

**Authors:** Vivek C. Velaga, Angus Cook, Kirsten Auret, Tom Jenkins, Geoff Thomas, Samar M. Aoun

**Affiliations:** 1Perron Institute for Neurological and Translational Science, 8 Verdun St, Nedlands, WA 6009, Australia; vivek.velaga@perron.uwa.edu.au; 2School of Population and Global Health, University of Western Australia, Clifton Street Building, Clifton Street, Nedlands, WA 6009, Australia; angus.cook@uwa.edu.au; 3Rural Clinical School of Western Australia, University of Western Australia, Building M701/31 Stirling Terrace, Albany, WA 6330, Australia; kirsten.auret@rcswa.edu.au; 4St John of God Midland Hospital, 1 Clayton Street, Midland, WA 6056, Australia; tom.jenkins@sjog.org.au; 5Thomas MND Research Group, 48 Grevillea Way, Blackwood, SA 5051, Australia; gtps@bigpond.net.au; 6Medical School, University of Western Australia, 8 Verdun St, Nedlands, WA 6009, Australia

**Keywords:** motor neurone disease, palliative care, end-of-life care, public health approach, family carers, bereaved, service provision, Compassionate Communities Connectors

## Abstract

Although the progressive clinical trajectory of motor neurone disease (MND) is widely understood, multiple challenges remain preventing optimal end-of-life care for this population with unique needs from the patient, carer and service provider perspectives. This paper reports on the experiences, gaps in service and unmet needs of MND patients and family carers and explores public health palliative care approaches that would facilitate coordinated and integrated care to respond to their changing needs. This is a qualitative study of responses to questions in an online consumer survey (353 respondents) in Western Australia (2020), focusing on a subset of 29 current and bereaved carers of people with MND who have used health services in the last five years. The analysis identified themes, highlighting the insufficient integration of services across health and social care; poor and unequal access to coordinated palliative care; significant gaps in the knowledge base of the workforce and a failure to meet the consumer expectations of person-centred care. For palliative care to be accessible to those living with MND and other under-served conditions, there needs to be a shift to more comprehensive, inclusive and sustainable options, such as the public health approach to palliative/end-of-life care that engages the assets of local communities in partnership with health services, one example being the “Compassionate Communities Connectors” model of care. Further considerations include advocacy for policy changes, fostering partnerships and developing indicators for evaluating the impact of the proposed models of care. The end result is not only better care but substantial savings for the health system.

## 1. Introduction

Motor neurone disease/amyotrophic lateral sclerosis (MND/ALS) is an incurable, fatal neurodegenerative disease typically involving rapid, progressive wasting, weakness and paralysis. Its cause is largely unknown, except for a small percentage of affected people with identifiable genetic abnormalities. The prevalence of MND is around 7 in 100,000, with the time between the onset of symptoms and death averaging 2–3 years [1,2]. With no truly effective treatments available to prevent disease progression, noting that Riluzole only prolongs survival by a median of 2–3 months [3], all cases are fatal. Respiratory failure and pneumonia are the most common modes of death. In Australia, there are currently about 2000 people and their families living with the disease [4]. An economic analysis estimated the total cost of MND to be $AUD1.13 million per person diagnosed, and the total annual cost of MND to the Australian community to be $AUD2.37 billion [5]. These costs were considerably higher than for other diseases mentioned in the report, primarily due to the costs associated with premature mortality and the disability burden to the individual, their families and carers, including absence from the workforce.

Despite the inexorable mortality associated with MND from the time of diagnosis, end-of-life outcomes are reported to be poor for patients with MND and their family carers. People with MND, their carers and health professionals encounter many complex barriers in receiving and providing high-quality end-of-life care (EOLC). The patients’ rapidly changing symptoms, requiring multidisciplinary approaches and specialised input, often affect their ability to participate in their own care decisions. Challenges include high and increasing levels of disability affecting physical independence, communication, cognition and behaviour [6]. These often disrupt patients with MND in terms of their social participation, autonomy and decision-making capacity [7]. Additionally, significant disability makes it increasingly difficult to travel for specialist review and to remain cared for at home. The implications are a loss of control as well as a poor understanding of what the future holds. Patients with MND fear a distressing death, with thoughts of respiratory distress, choking, communication loss, physical helplessness, dependency on others, loss of dignity and being a burden, contributing to an increased risk of suicide [2].

For families and carers, the challenges to achieving high quality EOLC include high levels of carer burden, financial and psychological distress, a need for education and training to meet complex care needs and monitoring of symptoms and limited respite options [8,9,10]. Outcomes are likely to be worse for families who are already at a disadvantage in terms of their ability to access and navigate the health system [11] because of isolation, language barriers and other factors. In addition, formal post-bereavement care is often fragmented or even non-existent [12,13].

For health professionals and service providers, there are various barriers to providing EOLC that is fully integrated with other services that the patient and family may already receive. Health professionals may lack a shared understanding of the patient’s and family’s goals and needs for care: this may stem from the limited knowledge and skills within both the general MND and EOLC workforces. Moreover, this lack of experience may result in service providers being uncomfortable with discussing end-of-life issues [14].

A recent study on consumer experiences in palliative care in Western Australia highlighted the differences in the quality of EOLC between cancer conditions, labelled as ‘winners’, and non-cancer conditions, being the ‘losers’, with the latter including MND [15]. In particular, quality indicators relating to families and carers being supported lagged behind other indicators of service quality. For example, families reported that they were not being well supported before and after bereavement in terms of their emotional needs, their ability to discuss their worries and fears with the services, being able to obtain information when needed and feeling included in care decisions.

This paper focuses on the MND carer experience as reported in the non-cancer conditions cohort in the study by Aoun et al. [15] and highlights the lived experience of patients and families, through the voice of the consumer and their stories, which may explain this inequity in quality of care.

## 2. Objectives

The objectives of this paper are to:-Report on the experiences, gaps in service and unmet needs of patients and carers through the qualitative comments provided by current and bereaved carers of people with MND.-Explore public health palliative care approaches that would facilitate coordinated and integrated care to meet the needs of people with MND and their family carers.

## 3. Methods

### 3.1. Ethics Approval

Ethics approval was granted by La Trobe University Research Ethics Committee (HEC20232). The information sheet that accompanied the survey emphasised that participation was entirely voluntary. Returning the completed, anonymised, online survey was considered implied consent.

### 3.2. Study Design

This is a qualitative analysis of responses to open-ended free text questions that were asked as part of an online cross-sectional consumer survey. This used a tailored questionnaire for patients with a life-limiting illness and their family carers (both users and non-users of palliative care services) who were resident in Western Australia in 2020 and are using or have used health services in the last five years.

The survey contained sets of questions related to: patient and carer demographics; introduction to palliative care; experience in the ‘home’ setting; experience in the ‘hospice’ setting; experience in the ‘hospital’ setting; experience in the ‘nursing home’ setting; services and community support before bereavement; circumstances around end-of-life/death; services and community support after bereavement.

### 3.3. Recruitment and Data Collection

Data were collected in June–July 2020. The survey was made primarily available as an online survey using Redcap. Paper versions of the surveys were also provided on request. Recruitment was mainly through social media. Complementing the survey tool were additional documents for respondents to clarify several aspects of the survey, including definitions of several used terms in the field; a list of palliative care service providers in Western Australia categorised by setting; information on palliative care information and services; information on grief and bereavement, including how to contact services should the respondent become distressed completing the survey and a patient and carer information sheet.

The survey was promoted extensively via service providers and relevant social media pages. Data collection was over a short six-week period. An advance direct mail was sent to approximately 130 individuals across palliative care providers, other government agencies, consumer organisations and other not-for-profit organisations, such as the MND Association in WA. During the survey open-period a follow-up direct mail was sent to the original listing, plus major hospital executive staff, additional personal contacts and additional identified organizations. In total, around 300 individuals representing more than 90 organisations were contacted plus all Western Australia parliamentarians. Personal contact was made with the major churches, funeral providers, various clinicians and health providers.

### 3.4. Study Population

In total, 430 surveys were received, with only three as paper copies. Following data cleaning to remove surveys which were 70% incomplete or were duplicates or were from patients not residing in WA, the final number suitable for analysis was 353, with 71% of the total respondents being bereaved carers and 68% of the total respondents having used a palliative care service.

For this article, the study sub-set comprised current and bereaved adult carers of individuals diagnosed with MND.

### 3.5. Analysis

Qualitative responses were analysed using an inductive thematic analysis as described by Braun and Clarke [16] and visualised in Thomas [17]. This approach allowed for a range of themes to be developed based on their prominence and prevalence across the survey results [16,17]. The survey responses were reviewed by three members of the study team (VV, AC and SA) to the extent of clearly discerning themes and meanings [16]. The software NVivo was used to assist with creating and keeping track of categories, themes, meaning and relevant quotes. Firstly, segments of text that directly commented on practices within services and among clinicians were identified as codes. These codes were then merged and refined based on similarities, which resulted in potential themes [16]. These themes were reviewed repeatedly to determine whether the code groupings were appropriate or if more appropriate groupings could be made [16]. The emerging interpretations were refined throughout the analysis processes and aided by a comparison between the data and the existing literature, enabling a data-driven approach to interpretation. The authors are palliative care or MND care researchers and therefore brought both theoretical and practical perspectives to interpreting the data, enabling a rigorous approach to reflexivity. All differences were discussed and consensus achieved, indicating the trustworthiness of the coding scheme. To ensure rigour, the COREQ checklist was followed, as described by Tong et al. [18] for reporting qualitative research.

## 4. Results

### 4.1. Characteristics of Study Population

In total, 353 valid surveys were received for the original survey for all conditions. The results presented here relate to the subset of responses received from current and bereaved carers of people with MND (*n* = 29).

Of these, 29 were carer respondents, 19 were bereaved carers who had use palliative care services in the past five years, 3 were current carer users of palliative care, 5 were bereaved carers who had not used palliative care services in the past and 2 were current carer non-users of palliative care. Sixty-nine percent of the respondents were female (*n* = 20) and the average age was 60 years (range: 33–76 years).

### 4.2. Overview of Major Themes

A thematic analysis of the responses provided insights into the 29 carers’ experiences. Six themes were identified.

The following abbreviations are used for the quotes’ IDs: CBU = carer bereaved user of palliative care; CBNU = carer bereaved non-user; CCU = current carer user; CCNU = current carer non-user.

(i)Emotional contexts of MND diagnosis and care

The sustained and demanding engagement in EOLC for patients with MND was extremely challenging emotionally for many carers, family members or friends. The overwhelming majority of respondents were either partners or children of the patient and hence carried a deep investment in the patient’s wellbeing and care.

*The whole process was terrible from start to finish and resulted, I feel, in his death* [happening] *sooner than it should have occurred*, [leaving] *myself more traumatised than I otherwise would have been*.(CBNU3)

The emotions reported by carers were diverse, extending beyond the expected feelings of sadness, loss and grief. One participant (CBNU55) felt guilty about watching her father pass away slowly with little professional or informal care, while another participant felt guilty that she could not arrange for her husband to receive EOLC at home:

*My husband’s time in hospice was extremely distressing for him, myself and all the family* … *The doctors and social worker tried to force me into putting him into aged care*.(CBU38)

One of the major triggers for the distress for carers and patients related to a perceived lack of emotional support from the professionals involved in their care, at times resulting in the carers feeling disregarded.

*Whilst we initially felt very lucky and grateful to be offered help with services, he felt very stressed and devalued by a lot of what was offered and how it was offered… It made him feel traumatised and totally hopeless*. (CBNU3)

In one instance the carer reported that she and her husband were made to feel unwelcome and were upset by a social worker attempting to move the patient into hospice care:

*We had some very upsetting meetings with the social worker there who was very unprofessional and was more interested in our financial ability for him to stay there than anything else*. (CBU87)

One carer was left feeling as though he himself was a burden and hindrance to their care for the patient:

*I was always concerned that I/we were seen to (be) complaining for the sake of complaining, and the need to have care staff “onside” was always a constant tension*. (CBU47)

However, four respondents did report having a positive experience with the level and quality of care received, and provided comments such as:

*We were overwhelmed by the care we were given. […] There was always someone there for us no matter what day or time*. (CBU48)

(ii)Settings and practicalities of care provision

Many of the study participants reported that they played a significant and leading role in providing care for the person with MND. This often resulted in carers taking on increased responsibilities throughout the care period, which at times were a physical and emotional burden.

[My mother] *was run ragged looking after dad* [who was affected by MND]. *He couldn’t move his arms from the very early stages so he couldn’t get a drink, turn on a light, go to the toilet, feed himself* etc. *We had to do all of that for him and more. Months and months of it. My mum got little to no respite and the only way she could have received any respite was to put dad into hospital which he vehemently didn’t want*.(CBNU55)

In many cases, the progression of MND gradually stripped patients of much of their independence, leaving them almost entirely reliant on carers for daily tasks, including basic self-care (at times round-the-clock). In addition to these ongoing responsibilities, most carers were also involved in liaising with facilities, allied health professionals and other providers.

*I spent many days leading up to his death visiting aged care homes, time that should have been spent with my darling husband. We had to have family present to attend to his needs and feed him every day*. (CBU38)

Carers of patients whose MND affected their ability to speak noted the significant toll that this had on the patient’s care:

*The nurses would go into his room but didn’t understand him or know what he wanted so they got frustrated with him. I overheard the nurse telling him off because he tried to phone me to come back to the hospice. I was very disappointed with her reaction*. (CBU63)

A few respondents raised the need for respite for carers. However, in some cases people with MND indicated that they only wanted to be cared for by their own family members or partners, therefore making carers reluctant to seek the respite they may have needed. The daughter of a patient made the following comment about the informal care that her mother received while caring for her husband:

*The physical reprieve my mum got when I was there to take over for her for a few hours* [was most helpful]. *Though dad was demanding and often only wanted mum to care for him. Having friends visit to break up his day made a difference for us all*. (CBNU55)

(iii)The role of informal support networks

Family and friends of the patient and their family often served as the centre of the support network for people with MND. Participants in the study stated that practical help was often appreciated, and at times it was friends who more frequently provided this than family members. When asked about the most helpful forms of support they received, one respondent mentioned:

*Chatting to family and friends about my husband. Practical help organising* [the] *funeral* etc.(CBU8)

In many situations, carers viewed close friends as being people they could more easily turn to talk to, especially in those situations where other family members had stepped away from involvement with the patient.

*Friends helped me emotionally, family abandoned us*. (CBNU51)

Respondents from rural areas demonstrated higher levels of support from friends and family and spoke of the benefits of living in a rural community:

*That was a benefit to living* [in a] *rural* [location]. *The medical staff all knew the situation and they helped as best they could. … Care in the home was well outside the scope of our local hospital but they did it for us all the same*. …[The person with MND] *did not want to go to hospital let alone leave his town so our local guys and gals did all they could*.(CBNU55)

*The job my mum did needs recognising! Without her, dad would have received no care or would have been shoved in a* [metropolitan] *care facility out of sight out of mind. No one, absolutely no one, would have cared for him the way my mum did*.(CBNU55)

(iv)Service availability and access

The availability of beds in hospitals and aged care facilities was found to be an issue for several respondents. One respondent (CBU47) reported that, although these institutions often had basic facilities, many were *“not well set up to meet the needs of those diagnosed with MND, especially when this was accompanied by comorbidities.”* Two other carers recalled that the appropriate equipment needed for an MND patient, including mobility aids, was often difficult to access in a timely manner. Others commented on the lack of access to palliative care.

*Our experience has been heartbreaking and we are still in the midst of it. My dad is being moved from hospital to hospital and finding palliative care is impossible*.(CCU26)

Some carers expressed their disapproval of the National Disability Insurance Scheme (NDIS), an Australian initiative which provides funding to people with disabilities under 65 years of age and connects them to providers and community groups. Respondents noted the delays they faced in receiving support and in having plans approved, as well as the lack of flexibility in care towards the end of the patient’s life:

*The NDIS process does not allow for quickly changing circumstances with conditions such as MND*.(CBU87)

*It took three months to get the first draft plan from NDIS. This arrived the day after his respiratory failure. It took three months. So he died waiting for the plan, let alone the services and care he needed*.(CBNU3)

A few respondents mentioned the Motor Neurone Disease Association of Western Australia as a service that helped care for people with MND through the provision of support services and staff training.

*MNDAWA* [the Motor Neurone Disease Association of Western Australia] *were very supportive and provided a community nurse who called in to see my husband whenever it was necessary*.(CBU8)

The central role of a community-based palliative care service, a leading provider of in-home care in mainly metropolitan Western Australia, was commonly mentioned in the responses, with many carers expressing gratitude for the role of this organisation in providing care for the person with MND.

[Service] *staff were always very caring, professional, and respectful of my husband’s needs and my situation*.(CBU17)

However, there was mixed feedback about the support provided by other external providers:

*Some* [externally employed] *carers and nurses were more compassionate and conscientious than others. There didn’t seem to be a standard across the board. It all came down to personality. …Some were amazing and some were awful.*(CBU17)

Rural respondents commented on their difficulties with accessing multiple services through their end-of-life journey. It was reported that local hospitals lacked doctors, nurses and the other resources needed to provide optimal care for people with MND.

*Our local doctor and hospital helped as best they could but* […had…] *limited resources to do the job they do, let alone make house calls. Allied health was limited and sporadic as they had to travel to the town. The doctor was often away.*(CBNU55)

One carer spoke of her father’s specialist being in the capital city, a 16 h return journey that her father would not be able to complete as he had become bedridden.

*Early on, we’d tried to find help. Living rural made that almost impossible. People shouldn’t be forced to go metro away from their homes and loved ones, especially while dealing with a diagnosis like MND… By the time allied health installed a wheelchair ramp my dad was unable to use the wheelchair. His progression was fast, but their response was also slow*.(CBNU55)

(v)Professional knowledge and service coordination

Some respondents recollected that some staff were not familiar with MND patients, which impacted their ability to provide these patients with an adequate standard of care.

*The nurses were not used to having someone with MND…. Some of the nurses were lovely and apologising for not having the training they needed, I would have to help them hoist him. …Staff said they were used to cancer patients who were easier to look after*.(CBU38)

One carer (CBU19) stated that *“the main issue is finding someone who has a real knowledge of MND and how quickly things change”*. Many respondents encountered issues when navigating the health system to organise care.

*Our care was chaotic. No one knew who was doing what, no coordination…This started from the diagnosis of MND* […] *and continued from his discharge from a private hospital without an adequate home care plan, incorrect referrals to inappropriate services and then totally uncoordinated care within our town and our home*.(CBNU3)

Often participants reported that this perceived lack of coordination between professionals resulted in increased responsibilities for the carers, who consequently felt that the amount of meaningful time they could spend with the patient was being compromised.

*Much of my time was spent trying to wrangle specialists and appointments whilst trying to deal with caring for my husband in the face of his terminal illness diagnosis… three months of stress and trauma trying to get the right care and support services in place instead of worthwhile time together. …The process, not his illness, exhausted us both and I believe took him before time*.(CBNU3)

Community MND and nursing services were often appreciated by many respondents as the first, and sometimes only, provider to refer patients to palliative care services for home care or hospice care. Conversely, one respondent (CBU48) recalled that the *“consulting specialist (was) not clear as to the correct pathways to access palliative care”.*

Finally, many carers described feeling emotionally taxed and distrustful of the health system during their journey with a patient with MND.

*The system is broken and needs to be fixed. MND is a diagnosis that needs its own system to help understand the impact*. […] *The systems are very flawed and for a 69-year-old to be forced into a nursing home is terrible when he wants to be in his own home*.(CCU26)

(vi)Managing Transitions to Terminal Care

Dying at home was highly valued by many patients and their family members, with one carer (CBU19) stating that *“my greatest achievement was in supporting and enabling my husband to die at home”.* Many family members sacrificed much of their own independence to provide for their loved ones, to ensure a death at home in alignment with the wishes of the patient.

*He wanted to be at home. He had the right to be at home, but that meant my mum had to quit work and be his sole carer. She had to be in his earshot 24/7 as all he could do was talk. By the last few months even that was getting harder for him. …She did it all for him because no one else could or would, without sending him hundreds and hundreds of kilometres from where he wanted to be*.(CBNU55)

The complexity in caring for an MND patient towards the end of life appeared to impact family members, especially regarding making difficult decisions involving patient care.

*He didn’t want to go on morphine but couldn’t handle the pain. The morphine made him comatose; he didn’t like that. He very clearly told me he wasn’t ready. That said, he was very very sick and not helping him with pain relief would have been cruel but my mum, to this day, blames herself for dad’s death. She called the hospital regarding his pain and they started the morphine. She tells me she killed him. It’s awful*.(CBNU55)

A recurring theme across the comments on EOLC was a desire to maintain a sense of control through this final phase of the condition:

*It was relatively quick and my husband was in control to the end. Something that was important when you lose control of everything else*.(CBU19)

Several respondents felt that the staff in care settings were not consistent in conveying updates on the status and wellbeing of patients, leaving family members unprepared when death occurred.

*I sat by my husband’s bedside for hours on the day he passed away. I thought he was just really sleepy, a nurse walks in and said, “he’s gone”. Nobody bothered to say he is slowly dying, they must have known. I could have called family for some support*.(CBU38)

*I would have liked to be with my husband during his final hours and when he passed. I think it was quite sudden and unexpected. I did receive a couple of phone calls during my husband’s last days but was not made fully aware of my husband’s deterioration*.(CBU8)

## 5. Discussion

This study evaluated first-hand experiences from the carers of people with MND and highlighted numerous issues directly impacting on the EOLC of this population. For most respondents, the health care system seemed to be poorly aligned with the needs of people with MND, and there was considerable scope for improvement in the palliative care and overall management of this complex condition. As discussed, commonly reported issues in the context of MND related to the impact of caring on carers and the support they received, professional training and health system coordination, service accessibility, preparation for death and respite and bereavement support for carers.

A significant number of respondents felt that health care providers were not sufficiently trained or experienced in the management of MND. In some cases, health professionals explicitly stated that they lacked the appropriate training and could not provide adequate support to the patient and their family. These observations have been raised in previous literature [11,19,20,21,22], including reports to government bodies and MND associations [23,24]. It has also been noted that professionals who lack expertise in MND management may feel overwhelmed by the task [25] and are less likely to fully anticipate the specific needs of these patients [21]. This highlights the need for doctors at clinics and support staff in MND associations to work closely together in developing and executing care plans.

Despite the majority of respondents using specialist palliative care services, they stated that they did not receive appropriate levels of empathy from their health care providers, a finding which is also consistent with other published accounts by MND patients and carers [11,26,27]. Shortcomings in professional support leads to increased psychological distress for carers [28]; carers and patients who feel they receive minimal empathy and support are likely to feel abandoned and frustrated. Their resulting loss of faith in the health system also reduces the likelihood that carers will seek the appropriate support when needed [20,26]. These carer observations indicate a potential need for further and specific training in MND for various professions, including nurses, allied health providers and medical professionals, or greater coordination with those who have this training [29,30].

The participants in our study also reported that, in the context of MND management, the coordination across the wider health system was relatively poor. For example, carers reported that some professionals did not recognise the appropriate points of care at which to refer patients to palliative care services. The Department of Health and Human Services [24] suggest that this issue may relate to the relative rarity of MND in many practices, and to the fact that the rapid progression of the disease may make the timely coordination of care difficult when shared between numerous providers. Furthermore, the role of palliative medicine in people with progressive neurological conditions is less fully established than for conditions such as cancer, and many patients with MND remain unaware of the importance and value of palliative care in their treatment [31,32,33].

The complex nature of the palliative care sector, which can be difficult to navigate, can also deter patients, families and carers from seeking palliative care, or may leave them at a loss regarding how to best access and arrange the support and professional help they need [34]. The carers of patients with MND in this study and others have reported that they find themselves lost in a “service provision maze”, and do not know who to approach for support when needed [10,23]. This issue—which has also been identified in other centres [26]—may in part be attributable to the complex nature of MND care, at times requiring the involvement of up to 22 different disciplines [35,36]. Poor system coordination may also be a product of inadequate communication between stakeholders and service providers to constructively work together to care for patients with MND [24]. In the current study it was noted that these challenges were more prominent in rural and regional areas, where high-quality staff and services were not as readily available and care was less coordinated [11].

A number of respondents indicated that they were not fully prepared for the death of the person with MND, and that the final stage seemed to occur without sufficient warning. In some cases, this was attributed to health providers failing to acknowledge or make the carers aware that the person with MND had entered the terminal phase. Such concerns around the time of death have also been identified by carers in other surveys [10,20,22,37]. A need for the improved management of bereavement was also evident, with some carers stating they were left unsupported and alone during the bereavement period. This may arise when the supporting services in place during the patient’s illness journey are suddenly withdrawn upon their death, leaving the carer feeling abandoned at the time they are starting to process grief and may need more support [11,19,23,27].

### 5.1. Limitations

There were several potential limitations in this study, the first being the limited time that was available for data collection. This study was carried out using input from carers in Western Australia only, limiting the generalisability of its conclusions. Carers with negative experiences may have been more motivated to share their experiences than those with positive experiences, thereby potentially skewing results. The method of data collection may not have provided sufficient opportunity for the respondents to share their entire perspectives and experiences. As a result, the findings may have been influenced more by experiences with greater emotional weighting, whereas less extreme experiences may not have been shared. Given that these results were generated using a qualitative methodology, they may be overly representative of those who shared multiple quotes with extensive detail. However, the findings are consistent with those reported in a national MND EOLC and bereavement survey where the sample was more representative of the MND bereaved population [12,13].

### 5.2. Necessity for Shifting Directions—Public Health Approaches to Palliative and EOL Care for MND

The responses by many of the carers in this analysis indicate that there is currently insufficient integration of services across health and social care, poor access to a coordinated palliative approach to care, significant gaps in the knowledge base of the workforce and a failure to meet the expectations of person-centred care. Public health approaches to EOLC may have the potential to enhance the integration of services and provide a comprehensive approach that engages the assets of local communities. Moreover, they offer frameworks in which partnerships can be developed with patient communities with distinctive end-of-life needs, such as those with non-cancer conditions, and, thus, provide a more inclusive approach to EOLC [38].

Planning for EOLC for people living with MND starts at the time of diagnosis, continuing through to bereavement support for family members [12,36,39]. An early intervention approach is needed, rather than reactive ‘last days’ service provision [40]. A common perception is that palliative care focuses on the last months and weeks of life, whereas it should begin at the time of the diagnosis and be an integral part of care plans. However, in much of the Western world, referral to palliative care mostly occurs late in the person’s life where the aim of providing excellent assessment and holistic care to improve quality of life may have less time to meaningfully impact on outcomes for patients and families [41].

The results from an Australian national bereavement survey [42] report that under the prevailing service provision model, more people with cancer (64%) received palliative care relative to other non-malignant illnesses (10.4% heart disease, 8.1% dementia, 7.7% organ failure, 4% lung disease). These patients with non-malignant diseases were under-represented in palliative care, with a median period of only one month in the service [42].

A palliative approach to care, on the other hand, places less emphasis on linking care provision to prognosis, but promotes early interventions aimed at having conversations with patients and their family members about their goals of care, comfort measures, needs and wishes [36,40]. A palliative approach needs to be integrated into the care plan for people with MND to optimise their quality of life by relieving symptoms, providing emotional, psychological and spiritual support pre-bereavement, minimising the barriers to a good death and supporting the family post-bereavement. These outcomes require the vital role of family carers and friendship networks, the involvement of MND associations, the education and training of general health and community care practitioners and constructive connections between informal caring networks and formal networks [36].

This approach is illustrated in Figure 1 [11] and recognises the input of the networks around the patient/family (circles of care) [43]. The ‘inner’ and ‘outer’ circles of care—being the family, neighbourhood and friendship supports—are the main foundation of resilient networks caring for people at home. One successful model of care is the “Compassionate Communities Connectors Program” where trained volunteers enhanced patients’ supportive social networks to deliver the practical and social support needed at home, alongside palliative care services, demonstrating formal and informal networks working together. The program outcomes included a significant improvement in social connectedness and a reduction in social isolation, less hospital admissions and shorter hospital lengths of stay. The program has been integrated into the standard practice of the health service [38,44]. With most MND care being provided at home, this program is particularly valuable for this population.

The overall public health approach seeks to achieve the integration of the disability, health and aged care sectors (tertiary, primary and community services, specialist palliative care, generalist palliative care, disease specific clinics and primary and allied health care services). The enablers for this integration include digital and assistive technologies, telehealth, advance care planning, education and training, not for profit organisations, such as the not-for-profit MND associations and a Compassionate Communities approach to care.

Frequently, the solution is seen as providing more resources. What this model advocates for is the more effective use and coordination of what is already available, which is managed through a patient/family centred approach to ensure that MND patients’ changing needs are understood and there is a coordinated approach to meeting them. The MND associations are best positioned to deliver this and have a central role in driving the Compassionate Communities approach to care.

## 6. Conclusions

For palliative care to be accessible to everyone and everywhere, a shift in direction is needed towards more comprehensive, inclusive and sustainable options for supporting positive end-of-life experiences for dying people, their carers and the community more broadly [45]. Many Australians currently die in a way and a place that is not reflective of their values or their choices and their end-of-life journey is disrupted by preventable or unnecessary admissions to hospital, with patients experiencing confusion and a loss of control and autonomy [46]. International research indicates that a solely clinical model of palliative care (mainly focusing on symptom management) is inadequate to address the multiple comorbidities and access issues that are characteristic of modern palliative care. Therefore, if palliative care is to successfully address the challenges of unequal access, coordination and continuity of care, and the very limited way in which health services think about EOLC, new practice models need to be identified, debated and tested. The application of these inclusive approaches, such as the public health approach to care, are of particular relevance to MND EOLC. A palliative care approach is an essential component of the shifting directions through the public health approaches to care.

Further work in this field needs to focus on: (i) advocating for policy changes and increased funding to support the implementation of comprehensive and inclusive palliative care models; (ii) fostering partnerships between healthcare providers, MND associations and community organizations to enhance collaboration and share resources in delivering quality end-of-life care; (iii) collaborating with policymakers and healthcare administrators to develop standardized metrics and quality indicators for evaluating the effectiveness and impact of palliative care services and the proposed approaches to care in the MND population. To this end, one Australian initiative, MINDAUS, a national patient centred registry, is driving advocacy, policy, collaborative service and research evidence to improve MND care [47].

## Figures and Tables

**Figure 1 brainsci-13-00920-f001:**
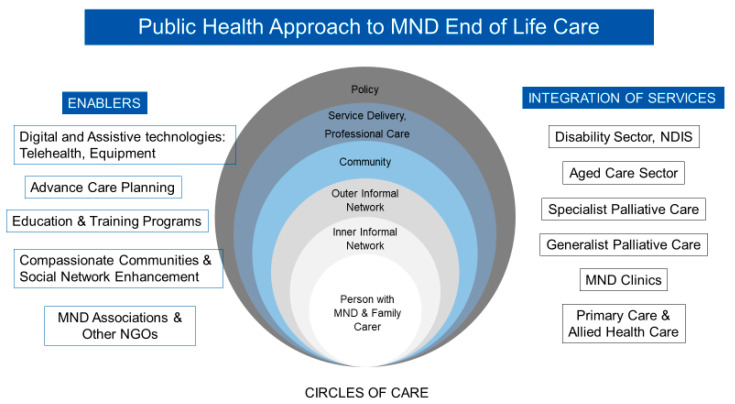
The public health approach to MND end-of-life care [11].

## Data Availability

Ethical approval precludes the data being used for another purpose or being provided to researchers who have not signed the appropriate confidentiality agreement. Specifically, the ethical approval specifies that all results are in aggregate form to maintain confidentiality and privacy and precludes individual-level data being made publicly available. All aggregate data for this study are freely available and included in the paper. Interested and qualified researchers may send requests for additional data to Samar Aoun at samar.aoun@perron.uwa.edu.au.

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
