# Peer review of "Palliative and End-of-Life Care for People Living with Motor Neurone Disease: Ongoing Challenges and Necessity for Shifting Directions"

_brainsci, 2023, doi:10.3390/brainsci13060920_

Round 1

Reviewer 1 Report

Comments and Suggestions for Authors

Congratulations on your work and contribution to the field of palliative care, particularly concerning individuals with Motor Neuron Disease (MND). Your research highlights the critical need for comprehensive, inclusive, and sustainable approaches to end-of-life care for dying individuals, their caregivers, and the broader community.

The abstract summarizes the key points of the study. The objectives of the paper are clearly stated. The abstract also mentions the methodology used.

Suggestions to improve:
State the significance: Please, clearly state why the study is important and how it addresses existing gaps in end-of-life care for individuals with motor neurone disease.

Expand on the methodology: Provide more detail on the qualitative study design and survey used in the research. Mention the number of participants and any specific criteria used for participant selection.

Emphasize the implications: How can addressing the identified challenges and gaps improve end-of-life care for individuals with motor neurone disease?

Background:
Explain why studying end-of-life outcomes and experiences is important for patients with motor neurone disease (MND). Highlight the unique challenges and needs of individuals with MND that differentiate them from other conditions.

Provide additional context: Expand the existing knowledge of end-of-life care for patients with MND by referencing previous research or studies. This will help situate the authors' study.

Methods section:

Provide more details on the survey: Expand on the specific details of the online cross-sectional consumer survey, such as how participants were recruited and any specific inclusion or exclusion criteria.

Explain the steps taken in the thematic analysis, including how the themes were identified and how the study team reviewed the data. Provide more information on how the codes were merged and refined to develop the themes.

Mention whether data saturation was achieved, indicating whether new themes continued emerging or if the identified themes covered most of the responses.

Elaborate on the expertise and background of the three study team members involved in reviewing and analyzing the survey responses.

Consider adding a paragraph on data validation: Discuss any steps taken to enhance the reliability and validity of the findings. For example, mention if inter-rater reliability checks were conducted or if member checking was performed to validate the identified themes with participants.

Lincoln and Guba created stringent criteria in qualitative research, known as credibility, dependability, confirmability and transferability – incorporate that to provide rigour to your document, my suggestion.

Results:

For the overview of major themes, the text presents six identified themes related to carers' experiences. Each theme is accompanied by quotes from the participants, categorized by their role as carer bereaved users/non-users of palliative care and current carer users/non-users.

The information provided is comprehensive, well-organized, and effectively conveys the study's main findings.

Some other implications that can be added to improve the conclusion (just suggestions, author decide):

·       Advocate for policy changes and increased funding to support the implementation of comprehensive and inclusive palliative care models.

·       Foster partnerships between healthcare providers, MND associations, and community organizations to enhance collaboration and share resources in delivering quality end-of-life care.

·       Collaborate with policymakers and healthcare administrators to develop standardized metrics and quality indicators for evaluating the effectiveness and impact of palliative care services in the MND population.

Reviewer 2 Report

Comments and Suggestions for Authors

A racional review of MND patients carer problems. It is mostly narrative, which emphasizes problems of individual carers and patients. But the reading is very attractive - such patients and carers are coming and to help them this can make problems. For a scientific paper - there are no facts, no tables, no statistics. Only 11 of 46 references are really recent. But discussion and conclusions are logical and inspiring. Concentration on palliative care and nurses education for end of life - this is necessary.
